# The Enantioselective Potential of NicoShell and TeicoShell Columns for Basic Pharmaceuticals and Forensic Drugs in Sub/Supercritical Fluid Chromatography

**DOI:** 10.3390/molecules28031202

**Published:** 2023-01-26

**Authors:** Denisa Folprechtová, Martin G. Schmid, Daniel W. Armstrong, Květa Kalíková

**Affiliations:** 1Department of Physical and Macromolecular Chemistry, Faculty of Science, Charles University, Hlavova 8, 12843 Prague, Czech Republic; 2Department of Pharmaceutical Chemistry, Institute of Pharmaceutical Sciences, University of Graz, 8010 Graz, Austria; 3Department of Chemistry and Biochemistry, University of Texas at Arlington, Arlington, TX 76016, USA

**Keywords:** NicoShell, TeicoShell, macrocyclic glycopeptides, superficially porous particles, enantioseparation, basic bioactive compounds, SFC

## Abstract

The enantioselective potential of two macrocyclic glycopeptide-based chiral stationary phases for analysis of 28 structurally diverse biologically active compounds such as derivatives of pyrovalerone, ketamine, cathinone, and other representatives of psychostimulants and antidepressants was evaluated in sub/supercritical fluid chromatography. The chiral selectors immobilized on 2.7 μm superficially porous particles were teicoplanin (TeicoShell column) and modified macrocyclic glycopeptide (NicoShell column). The influence of the organic modifier and different mobile phase additives on the retention and enantioresolution were investigated. The obtained results confirmed that the mobile phase additives, especially water as a single additive or in combination with basic and acidic additives, improve peak shape and enhance enantioresolution. In addition, the effect of temperature was evaluated to optimize the enantioseparation process. Both columns exhibited comparable enantioselectivity, approximately 90% of the compounds tested were enantioseparated, and 30% out of them were baseline enantioresolved under the tested conditions. The complementary enantioselectivity of the macrocyclic glycopeptide-based chiral stationary phases was emphasized. This work can be useful for the method development for the enantioseparation of basic biologically active compounds of interest.

## 1. Introduction

Macrocyclic glycopeptides (MGs) are broadly utilized as chiral selectors in liquid chromatography (LC) [1] and capillary electrophoresis [2] and are successfully employed in sub/supercritical fluid chromatography (SFC) [3,4]. Their complex structure and multiple functional groups allow enantiorecognition of a broad range of compounds [5]. The stereogenic centers of MGs are surrounded by numerous potential sites for various interactions (e.g., hydrophobic, electrostatic, H-bonding, steric repulsion, dipole stacking, π–π interactions) between the analyte and stationary phase that contribute to chiral recognition [6,7]. Furthermore, MG-based chiral stationary phases (CSPs) have structural similarities that frequently lead to complementary properties [8]. Hence, if partial enantioseparation is achieved with one column, there is a good chance for baseline enantioseparation with another [1,3]. Recently, MGs were bonded to superficially porous particles (SPPs) to enhance chromatographic performance by shortening the analysis time and improving efficiency and enantioresolution and enantioselectivity [3,9,10,11].

SFC has become a powerful technique in the pharmaceutical industry, mainly for the enantioseparation of chiral compounds [12,13,14,15,16]. In contrast to LC, low viscosity and high diffusivity of the mobile phase (MP) enable the performance of effective separations at a higher flow rate with a smaller pressure drop [17,18]. SFC, in combination with SPP-based columns, results in faster mass transfer kinetics, which improves enantioseparations affected by slow adsorption–desorption kinetics [19]. The composition of the MP also significantly impacts the enantioseparation performance. Modern SFC uses MP composed of CO_2_ with miscible organic modifiers such as methanol (MeOH), ethanol (EtOH), propane-2-ol (PrOH), acetonitrile (ACN) (often in combination with alcohol), and additives if needed [20]. The addition of organic modifiers increases the elution power of the MP and affects the retention/interaction mechanism, resulting in different selectivities [21]. MP additives are utilized for many purposes, including changing the apparent pH and polarity of the MP, enhancing analyte solubility, acting as an ion-pairing agent, suppressing ionization, and reducing unfavorable interactions [22,23]. Besides conventional acids and bases, water has been used recently as an MP additive [24]. Water in the MP significantly improves separation efficiencies and decreases retention times when using polar stationary phases such as teicoplanin- or vancomycin-based CSPs [25,26]. The main benefits of using water as an additive in MP include an increase in the range of polarities of separable analytes in SFC, e.g., peptides, new CSP selectivities, or an improvement in peak symmetry [19,27]. Furthermore, water is said to compete with the active sites and potentially shift the adsorption isotherm, causing peak shapes to switch from tailing to fronting [19,25]. Although the use of water as an additive is increasing, the fundamental mechanisms of its role have yet to be completely understood [19].

In the literature, several papers focused on the use of MG-based SPP packed columns in SFC have been published [3,6,25,27,28,29]. The Armstrong group demonstrated the broad applicability of SPP-functionalized MG-based CSPs, including vancomycin-based, teicoplanin-based, and NicoShell—the new MG-based column in SFC, for the first time. NicoShell utilizes a synthetically modified macrocyclic glycopeptide as a chiral selector and has shown broad enantioselectivity for basic compounds containing secondary and tertiary amines. Previously, we have shown the enantioselective potential of TeicoShell and VancoShell columns for ketamine and cathinone derivatives in SFC [6]. We have also demonstrated the high enantioseparation ability of the VancoShell column for new psychoactive compounds, including a large group of pyrovalerone derivatives in SFC [28]. Owing to the complementary behavior of these columns and the presence of secondary and tertiary amines in their structure, we sought to evaluate the enantioselective potential of TeicoShell and NicoShell columns for basic chiral biologically active compounds, namely, pyrovalerone, ketamine, cathinone derivatives, and other representative psychostimulants and antidepressants. We focused on a systematic study of MP composition, i.e., amount and type of organic modifier and additives (basic, acidic, mixed additives, and water), effects on retention, and enantioseparation of basic chiral compounds. The impact of separation temperature was also assessed. In this work, the difference in the enantioselective potential of these MG-based columns and their complementary behavior were evaluated and discussed. Another goal of this research was to evaluate the structure- or property-related behavior of the analytes on these MG-based CSPs in SFC and to generalize the results.

## 2. Results and Discussion

The effect of individual organic modifiers, i.e., MeOH, EtOH, PrOH, and ACN; individual additives, i.e., trifluoroacetic acid (TFA), acetic acid (AcA), diethylamine (DEA), triethylamine (TEA), isopropylamine (IPA), water, and their various combinations; and their amounts in the MPs on retention and enantioseparation of 28 analytes was investigated (structures of these compounds are displayed in Appendix A). Table 1 summarizes the best-obtained results (i.e., retention time of the first eluted enantiomer (*t*_R1_), enantioselectivity (*α*), and resolution (*R*)) for both MG-based CSPs. If baseline separation was achieved under several different conditions, the listed results were those that provided the shortest analysis time and *R* ≥ 1.5. Chromatographic data obtained for all analytes under various MP compositions tested on both CSPs are listed in Appendix A.

As shown in Figure 1, both MG-based columns showed identical and similar enantioselective potential for derivatives of cathinone and for derivatives of pyrovalerone, respectively. Approximately 90% of the compounds tested were enantioseparated (*α* ≥ 1.1) and 30% had resolution values higher than 1.5 on both NicoShell and TeicoShell columns under the tested conditions. Table 1 shows that some representatives of individual groups such as 4-Cl-PVP, 3,4-MD-PHP, 4-CDC, DXE, or chlorthalidone were baseline enantioseparated on one column, while on the other, only an indication of enantioseparation was seen under the same separation conditions. This complementary behavior of MG-based columns is illustrated in Figure 2 for chlorthalidone and M-PPP.

### 2.1. Effect of Organic Modifier

The influence of three organic modifiers, i.e., MeOH, EtOH, and PrOH, on the enantioseparation of basic chiral compounds was tested on both columns. For the NicoShell column, MeOH was the most successful organic modifier (see Appendix A). Higher retention, typically with enantioresolution loss, and worse peak symmetry were achieved using EtOH or PrOH in the MP (data not shown). Appendix A illustrates the effect of MeOH and EtOH used as organic modifiers on the enantioseparation of pyrovalerone 4-F-PV8 on the NicoShell column. In the case of the TeicoShell column, EtOH provided higher enantioresolution values for most tested analytes (Appendix A). A combination of EtOH and a small amount of PrOH was the most suitable for the enantioseparation of pyrovalerone derivatives, except for PV9 and PV10 (see Table 1). Using only PrOH as a modifier in the MP resulted in a loss of enantioseparation, higher retention, and peak broadening as on the NicoShell column. A mixture of MeOH and ACN was also tested. The presence of ACN only improved the enantioseparation of derivatives of ketamine. A higher enantioresolution value of these compounds was observed in the MP: CO_2_/MeOH/ACN/TFA/DEA (90/8/2/0.05/0.5, *v*/*v*/*v*/*v*/*v*) than in the same ACN-free MP composition (Appendix A).

### 2.2. Effect of MP Additives

The effect of acidic and basic additives, water, and their combination on the retention, enantioresolution, and enantioselectivity was investigated because MG-based CSPs did not yield satisfactory baseline enantioseparation in additive-free MPs. The NicoShell column was able to partly enantioseparate more than half of the tested compounds in the MP CO_2_/MeOH (80/20 *v*/*v*). In contrast, the TeicoShell column could not resolve most of the analytes in additive-free MPs (see Appendix A). Additives affect the polarity and pH of the MP, as well as the dissociation and protonation of solutes [13]. As MGs contain ionizable groups (even though the exact structure of the NicoShell CSP is not available), additives also influence the state of the CSP owing to the ionization of the chiral selector. Based on our previous works [6,28,30], mixed additives (TFA + IPA, AcA + TEA) were used during screening measurements as they have yielded better results than single basic or acidic additives in the MP. Using only a single basic additive had a negative impact on enantioseparation; retention was significantly reduced, and thus enantioresolution was lost on both columns. However, for compounds with higher retention in additive-free MP, such as *α*-PPP and M-PPP, using 0.1% TEA in the MP resulted in fast baseline enantioseparation with enhanced peak symmetry on the NicoShell column. Compared with additive-free MP, the use of 0.1% TFA resulted in a slight decrease in retention as well as a decrease in enantioresolution on the NicoShell column. As seen in Table 1, the best results for most compounds were obtained in the MP with a mixture of additives if the amount of the acidic additive was much higher than the basic additive. In the case of the TeicoShell column, a combination of TFA and TEA at the same ratio, and in many cases also with water, yielded the best results among the tested MPs (see Table 1). Using AcA instead of TFA resulted in lower retentivity and loss of enantioresolution of pyrovalerone and cathinone derivatives (Appendix A). The best combination of additives on the NicoShell column varied depending on the nature of the tested compound. In several cases, a small amount of water in the MP enhanced enantioseparation and peak symmetry. Especially those that could be partly enantioseparated on the NicoShell column in the additive-free MP, such as pyrovalerones 4-F-PV8 and 4-Cl-PVP or cathinones 4-CBC and 4-CIC. Using 1.25 vol% of water in the MP resulted in a baseline resolution of enantiomers of 4-Cl-PVP with significant improvement in peak symmetry. With a higher amount of water (1.50 vol%) in the MP, the resolution value slightly decreased, and the peaks began to front (see Figure 3). This behavior is related to the ability of water to change the adsorption isotherm, shifting peaks from tailing to fronting [19].

Likewise, water with AcA and TEA in the MP significantly improved resolution for some compounds, e.g., pyrovalerone 3,4-MD-PHP, cathinone 4-CDC, or ketamine DXE (see Figure 4a). In contrast, using TFA instead of AcA caused a reduction or complete loss of resolution (see Appendix A). Water as an additive in the MP also positively affected the enantioresolution of some compounds on the more polar TeicoShell column. Depending on the type of compound, the best results were obtained at concentrations of 0.25 and 0.5 vol% water with 0.2 vol% TFA and 0.2 vol% TEA in the MP (see Table 1). The effect of water on the enantioseparation of DXE and 2-MeO-ketamine in a mixture with acidic and basic additives in the MP on the NicoShell and TeicoShell columns is illustrated in Figure 4a,b, respectively.

### 2.3. Effect of Temperature

Various column temperatures (25–40 °C) were tested during the optimization of separation conditions. As shown in Table 1, the best results for most of the tested compounds were obtained on the TeicoShell column at 25 °C, particularly for MPs with water as an additive. Increasing the separation temperature in the systems with water as a MP additive decreased the retention and enantioresolution values for many compounds tested on the NicoShell column (see Appendix A). The separation temperature in the systems with the TeicoShell column did not affect the retention as much as on the NicoShell column. The effect of the separation temperature on the enantioresolution was different for individual compounds on the TeicoShell column (see Appendix A). It must be noted that the effect of temperature varies depending on the analyte and the MP composition used. Based on the dependencies of ln *α* and ln *t*_R1_ on 1/*T* for selected pyrovalerone shown in Appendix A, we can see a slight deviation from linearity on the NicoShell column. However, the range of experimental data is limited. This trend was observed for all compounds tested with the same MP. On the other hand, no trend was observed on the TeicoShell column and compounds showed different behavior under the tested temperatures (for illustration, see Appendix A). Separation temperatures of 40 °C resulted in higher enantioresolution in water-free MPs for some compounds, such as praziquantel, *N*-ethylketamine, *α*-PPP, or M-PPP, on both columns. As a result, the temperature values in Table 1 differ.

### 2.4. Enantioselectivity for Derivatives of Pyrovalerone

The NicoShell column exhibited excellent enantioselectivity for both *α*-PPP and M-PPP. The only difference between them is the -CH_3_ substituent at the para position of the benzene ring (see Appendix A for analyte structures). *α*-PPP and M-PPP were baseline resolved (*R* = 3.0, *R* = 3.3, respectively) in an MP of CO_2_/MeOH/TEA (90/10/0.1 *v*/*v*/*v*) within 3 min. In contrast, the TeicoShell column could only partly enantioresolve them with significantly longer analysis times. According to the obtained data (Appendix A), retention on both columns decreased with increasing length of the alkyl chain of a series of related compounds, such as *α*-PPP, *α*-PVP, PV9, and PV10. This behavior can be correlated with the polarity of individual pyrovalerone derivative expressed as log *P* values—see Table 1. However, the enantioresolution of these derivatives did not correlate with this trend. It changed depending on the MP used. The electron-donating substituent at the para position of the benzene ring positively affected the enantioresolution of the pyrovalerone derivatives on the TeicoShell column. For example, 4-MeO-*α*-PVP with the -OCH_3_ substituent was nearly baseline enantioseparated (*R* = 1.40), and 4-MPrC with a -CH_3_ substituent was successfully baseline resolved (*R* = 1.53). Conversely, electron-withdrawing substituents (-F, -Cl) at 4-F-PVP and 4-Cl-PVP decreased enantioseparation on the TeicoShell column in the same MP (see Figure 5). Both *t*_R_ and *R* values decreased with an increase in the electronegativity of the halogen atom. Overall, the retention and resolution decreased for compounds containing electron-withdrawing substituents compared with those with electron-donating substituents. The only exception was 4-MeO-*α*-PVP, which had higher retention and lower resolution than 4-MPrC. The influence of the analyte structure on the retention and enantioseparation of selected pyrovalerone derivatives is shown in Figure 5.

The NicoShell column showed similar trends for the retention of these compounds. Enantiomers of 4-Cl-PVP were successfully baseline resolved in the MP CO_2_/MeOH/H_2_O (80/20/1.0, *v*/*v*/*v*), where 4-F-PVP, 4-MPrC, and 4-MeO-*α*-PVP were partly enantioseparated in various MPs containing different mixed additives—see Table 1. Concerning naphyrone (naphthalene substituent) and TH-PVP (tetrahydronaphthalene substituent), both columns could enantioseparate these compounds to some extent, but baseline enantioseparation was observed only for TH-PVP on the TeicoShell column. On both columns, naphyrone was more retained than TH-PVP with the same MP, most likely because of the strong π–π stacking with MG-based selectors. Different behavior of these columns was observed for the enantioseparation of 5-DBFPV. The NicoShell column partly resolved this compound, whereas baseline enantioseparation was achieved on the TeicoShell column. In contrast, the NicoShell column exhibited high enantioselectivity for 3,4-MD-PHP, whereas the TeicoShell column showed no sign of enantioseparation (see Table 1 and Appendix A for analyte structures).

In summary, the NicoShell column exhibited higher enantioselectivity for individual representatives of tested groups, such as 4-Cl-PVP, *α*-PPP, M-PPP, and 3,4-MD-PHP. On the other hand, the TeicoShell column was a better choice for enantioseparation of 4-MPrC, TH-PVP, and 5-DBFPV.

### 2.5. Enantioselectivity for Derivatives of Cathinone

Each column exhibited different enantioselectivity and retentivity for individual chlorine-containing cathinones, as shown in Table 1 and Figure 6.

The NicoShell column could baseline enantioseparate compounds 4-CBC (a secondary amine) and 4-CDC (a tertiary amine). However, compound 4-CIC (containing secondary amine) was only partially enantioresolved, likely because of steric hinderance from the bulky isopropyl side chain. Unlike the NicoShell column, the TeicoShell column baseline resolved the enantiomers of 4-CIC, but only partially for the enantiomers of 4-CBC. No enantioseparation was observed for 4-CDC containing a tertiary amine near the stereogenic center in any of the MPs tested on the TeicoShell column. The NicoShell and TeicoShell columns provided opposite retention trends of the cathinone derivatives. As shown in Figure 6, on the TeicoShell column, retention increased as the log *P* value of the given compounds decreased, whereas the reverse trend was found on the NicoShell column in all MPs tested (Appendix A). For illustration, the different MP compositions on each column were chosen with respect to the resolution of enantiomers.

### 2.6. Enantioselectivity for Derivatives of Ketamine

As shown in Table 1, the NicoShell and TeicoShell columns offered interesting results for the three ketamine derivatives tested. Both columns could baseline enantioseparate *N*-ethylketamine (with an electron-withdrawing -Cl group) under different MP compositions. The presence of the methoxy group as an electron-donating substituent on the benzene ring affected the enantioseparation on each column differently. The TeicoShell column could baseline resolve enantiomers of 2-MeO-ketamine in various MPs tested (see Appendix A) but, using the NicoShell column, only a slight indication of enantioseparation was achieved (MP with water as an additive) (see Appendix A). However, the NicoShell column could baseline resolve enantiomers of DXE (with no substituent) in the MP with 0.1% of AcA and 0.01% of TEA, and 0.5% of water as additives (Table 1), but with the TeicoShell column, only partial separations were found in various MPs tested (Table 1 and Appendix A). Unlike para-substituted pyrovalerones, higher retention was observed for DXE than 2-MeO-ketamine in all MPs tested on both columns (Appendix A).

### 2.7. Enantioselectivity for Others

Except for amphetamine and methamphetamine, the last group of tested analytes does not share a similar molecular structure, as shown in Appendix A. Therefore, any general discussion cannot be performed. Nevertheless, we want to show these results as they can be helpful to those who have to carry out enantioseparations of these active pharmacological compounds. At any MP used, neither column could enantioseparate amphetamine with a primary amine in the structure. On the TeicoShell column, there was only a slight indication of enantioseparation of methamphetamine. Partial enantioseparation of citalopram was achieved on the NicoShell column in MP composed of CO_2_/MeOH/DEA (90/10/0.1, *v*/*v*/*v*), while no enantioseparation of this antidepressant was obtained on the TeicoShell column. Surprisingly, the NicoShell and Teicoshell columns showed similar enantioselectivity for praziquantel, an anthelmintic drug. Partial enantioseparation of this compound was achieved under the same chromatographic conditions in MP: CO_2_/MeOH/AcA/TEA (90/10/0.5/0.01, *v*/*v*/*v*/*v*) on the NicoShell column and CO_2_/MeOH/AcA/TEA (90/10/0.2/0.02, *v*/*v*/*v*/*v*) on the TeicoShell column—Table 1. The TeicoShell column provided good enantiorecognition ability for chlorthalidone. This diuretic drug was baseline enantioseparated in almost all MPs tested (Appendix A). Conversely, only slight enantioseparation of chlorthalidone was observed on the NicoShell column—Table 1. Both TeicoShell and NicoShell columns showed excellent enantioselectivity for modafinil, a central nervous system stimulant used for “sleepiness”. This chiral drug has an asymmetric sulfoxide group and was baseline enantioseparated with almost all MPs tested on both columns (Appendix A). The chromatograms of fast enantioseparations of modafinil on both columns under different MP compositions are shown in Figure 7. Good peak symmetry values were achieved in both cases, i.e., *As*_1_ = 1.33, *As*_2_ = 1.38 on the NicoShell column, and *As*_1_ = 1.25, *As*_2_ = 1.20 on the TeicoShell column.

## 3. Materials and Methods

### 3.1. Chemicals and Reagents

Propane-2-ol (PrOH, for HPLC, ≥99.9%), acetonitrile (ACN, gradient grade, for HPLC, ≥99.9%), diethylamine (DEA, ≥99.5%), 2-propylamine (IPA, ≥99.5%), triethylamine (TEA, ≥99%), trifluoroacetic acid (TFA, 99%), and acetic acid (ReagentPlus^®^ > 99%) were supplied by Sigma–Aldrich (St. Louis, MO, USA). Methanol (MeOH, supergradient grade, for HPLC, ≥99.9%) and ethanol absolute (EtOH, gradient grade, for HPLC, >99.7%) were purchased from VWR International (Radnor, PA, USA). Water (Chromasolv Plus, for HPLC) was obtained from Honeywell (Charlotte, NC, USA). Pressurized liquid CO_2_ 4.5 grade (99.995%) was purchased from Messer (Prague, Czech Republic). Chiral compounds: pyrovalerone, cathinone, ketamine, and amphetamine derivatives were purchased from internet vendors and tested for identity prior to use. Modafinil was purchased from Carbosynth Ltd. (Compton, United Kingdom). Chlorthalidone, citalopram, and praziquantel were supplied by Sigma–Aldrich (St. Louis, MO, USA). The list of compounds used in this work with their structures is given in Appendix A. 

### 3.2. Chromatographic Conditions

SFC measurements were carried out on the Waters Acquity Ultra Performance Convergence Chromatography (UPC^2^) system. This instrument includes a binary solvent delivery pump compatible with mobile phase flow rates up to 4 mL min^−1^ and pressures up to 41.37 MPa, an autosampler with a partial loop volume injection system, a BP regulator, a column oven, and a photodiode array detector (Waters Corporation, Milford, CT, USA). The Empower 3 software was used for system control and data acquisition. Superficially porous particles (2.7 μm) bonded teicoplanin (TeicoShell column) and synthetically modified MG (NicoShell column) CSPs, both 100 × 2.1 mm I.D., were obtained from AZYP, LLC (Arlington, TX, USA). MPs were composed of CO_2_ and organic modifiers with the addition of different additives in various volume ratios. Additives were added to the organic modifier in appropriate amounts depending on modifier percentage in the MP. Screening chromatographic conditions were as follows: temperature of 40 °C; BP of 13.8 MPa; MP flow rate of 2.00 mL min^−1^; and UV detection at wavelengths of 220, 254 nm, and 280 nm. During the optimization procedure, various column temperatures (20–40 °C) and flow rates (1.00–2.00 mL min^−1^) were tested. The injected volume was 0.2–1.0 μL according to the detector response. The sample temperature was 10 °C. The dead time value was determined by the first system peak. All measurements were performed in triplicate. Stock solutions of individual analytes at a concentration of 0.5 mg mL^−1^ were prepared by dissolving samples in methanol and were filtered using a 0.2 μm PTFE filter. MarvinSketch 17.29 software was used to calculate the log *P* values of the tested analytes (see Table 1).

## 4. Conclusions

In this work, the enantioselective potential of TeicoShell and NicoShell columns for basic biologically active compounds in SFC was assessed. We examined the influence of the amount and type of organic modifiers and additives on the retention and enantioresolution of the compounds of interest. MeOH as an organic modifier was better choice for the NicoShell column, whereas EtOH or a mixture of EtOH/PrOH was preferred for the TeicoShell column. Moreover, the addition of water as a single additive or in mixture with basic and acidic additives improved enantioresolution and resulted in more efficient enantioseparations of many compounds. The influence of temperature varied with the analyte structure and the MP used. However, when using MP with water as an additive, a temperature of 25 °C resulted in significantly better enantioresolution.

As MGs have an analogous structure, both CSPs demonstrated comparable enantioselectivity in terms of the number of enantioseparated compounds, but the complementary behavior of these CSPs was observed in many cases. Both TeicoShell and NicoShell columns could enantioseparate 90% of compounds tested. The NicoShell column exhibited high enantioselectivity for individual representatives of tested groups, such as 4-Cl-PVP, *α*-PPP, M-PPP, 3,4-MD-PHP, 4-CBC, 4-CDC, and DXE. On the other hand, the TeicoShell column was the better choice for 4-MPrC, TH-PVP, 5-DBFPV, 4-CIC, or 2-MeO-ketamine. Both columns also offered fast and efficient enantioseparation of modafinil. The structural similarities of the individual analytes, i.e., different substituents or the length of the alkyl chain, and their enantioseparation behavior within the group were discussed.

As a result, the combination of these two MG-based CSPs provides a useful tool for the enantioseparation of a wide range of basic structurally distinct pharmaceutical and forensic drugs in SFC.

## Figures and Tables

**Figure 1 molecules-28-01202-f001:**
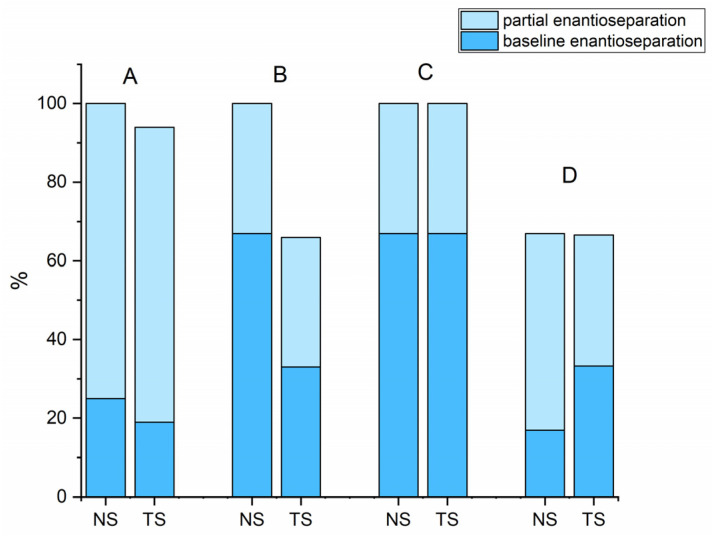
The overall success rate of baseline and partial enantioseparations for individual groups of basic biologically active compounds in different MPs; A—derivatives of pyrovalerone, B—derivatives of cathinones, C—derivatives of ketamines, D—others. Partial enantioseparation: *α* ≥ 1.1; baseline enantioseparation: *R* ≥ 1.5.

**Figure 2 molecules-28-01202-f002:**
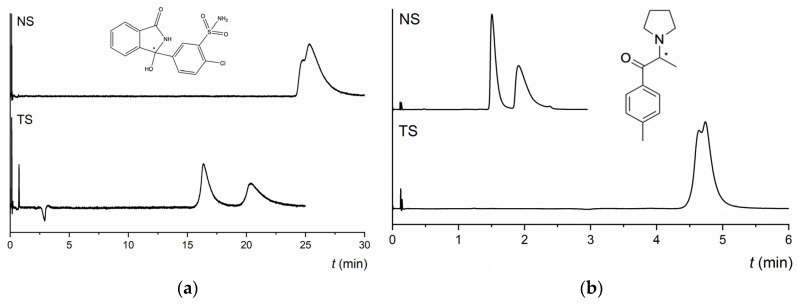
Chromatograms of enantioseparation of chlorthalidone (**a**) and M-PPP (**b**) on the NicoShell (NS) and TeicoShell (TS) columns. MP composition: CO_2_/MeOH/AcA/IPA (90/10/0.1/0.01, *v*/*v*/*v*/*v*) on both columns. SFC conditions: flow rate 2.00 mL min^−1^; column temperature 40 °C; BP 13.8 MPa; UV detection 254 nm. (*) in the structure indicates a stereogenic center.

**Figure 3 molecules-28-01202-f003:**
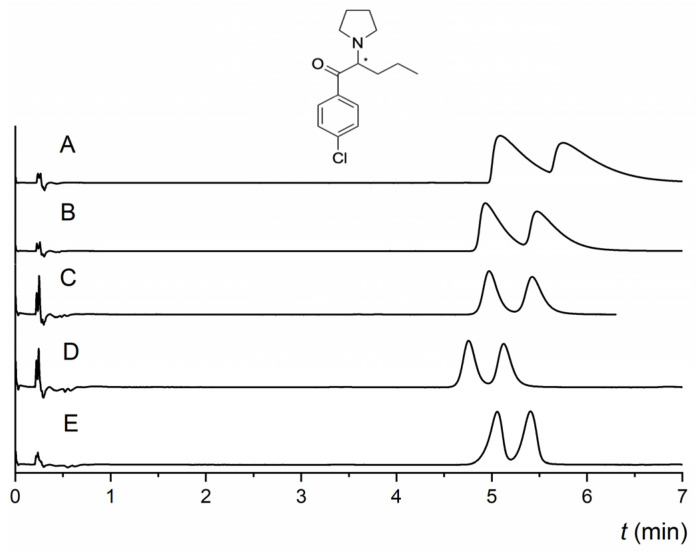
Effect of the amount of water as an additive on the enantioseparation of pyrovalerone (4-Cl-PVP) on the NicoShell column. MP composition: CO_2_/MeOH 80/20 (*v*/*v*) (A), CO_2_/MeOH/H_2_O 80/20/0.5 (*v*/*v*/*v*) (B), CO_2_/MeOH/H_2_O 80/20/1.0 (*v*/*v*/*v*) (C), CO_2_/MeOH/H_2_O 80/20/1.25 (*v*/*v*/*v*) (D), CO_2_/MeOH/H_2_O 80/20/1.5 (*v*/*v*/*v*) (E), flow rate 1.00 mL min^−1^, temperature 25 °C, BP 13.8 MPa, UV detection 254 nm. (*) in the structure indicates a stereogenic center.

**Figure 4 molecules-28-01202-f004:**
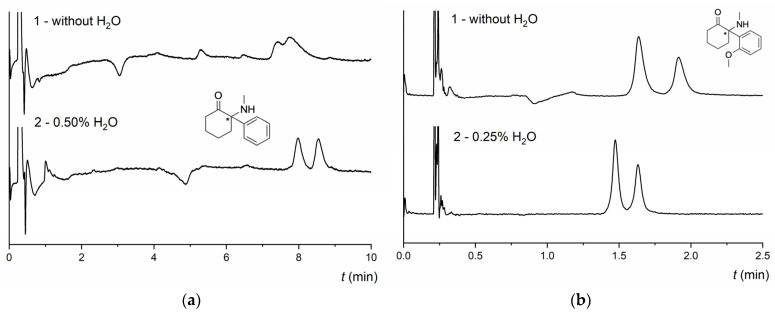
(**a**) Chromatogram of enantioseparation of DXE on the NicoShell column. Chromatographic conditions: MP CO_2_/MeOH/AcA/TEA (90/10/0.1/0.01, *v*/*v*/*v*/*v*) (1) and CO_2_/MeOH/H_2_O/AcA/TEA (90/10/0.5/0.1/0.01, *v*/*v*/*v*/*v*/*v*) (2); flow rate 1.00 mL min^−1^; column temperature 25 °C; BP 13.8 MPa; UV detection 210 nm. (**b**) Chromatogram of enantioseparation of 2-MeO-ketamine on the TeicoShell column. SFC conditions: MP CO_2_/EtOH/PrOH/TFA/TEA (85/12/3/0.2/0.2, *v*/*v*/*v*/*v*/*v*) (1) and CO_2_/EtOH/PrOH/H_2_O/TFA/TEA (85/12/3/0.25/0.2/0.2, *v*/*v*/*v*/*v*/*v*/*v*) (2); flow rate 1.20 mL min^−1^; column temperature 25 °C; BP 13.8 MPa; UV detection 280 nm. (*) in the structure indicates a stereogenic center.

**Figure 5 molecules-28-01202-f005:**
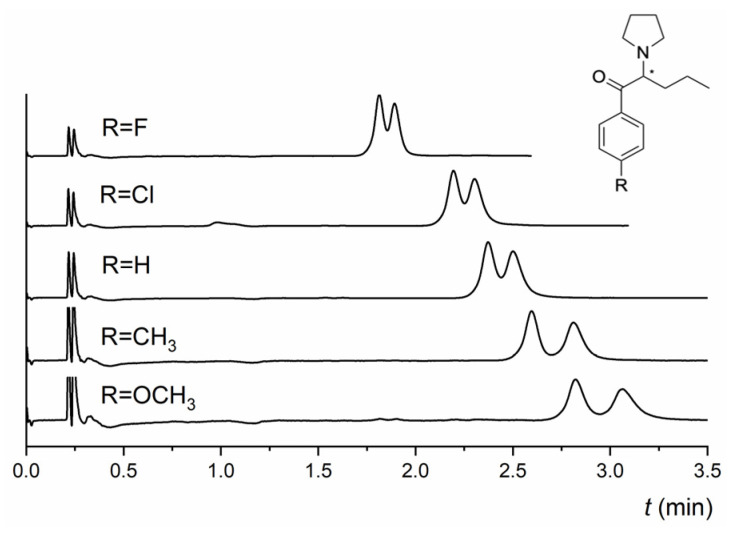
Influence of the analyte structure on the retention and enantioseparation of pyrovalerone derivatives on the TeicoShell column; 4-F-PVP, 4-Cl-PVP, *α*-PVP, 4-MPrC, and 4-MeO-*α*-PVP. MP composition: CO_2_/EtOH/PrOH/H_2_O/TFA/TEA (85/12/3/0.25/0.2/0.2, *v*/*v*/*v*/*v*/*v*/*v*). SFC conditions: flow rate 1.20 mL min^−1^; column temperature 25 °C; BP 13.8 MPa; UV detection 254 nm. (*) in the structure indicates a stereogenic center.

**Figure 6 molecules-28-01202-f006:**
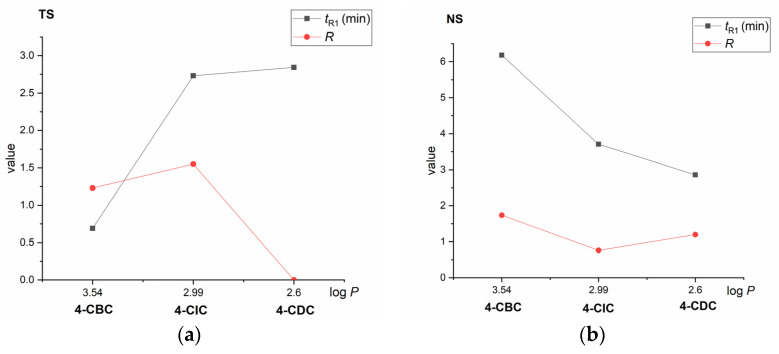
Dependencies of retention (*t*_R1_) and resolution values (*R*) on polarity (log *P*) of derivatives of cathinone on the TeicoShell (TS) (**a**) and NicoShell (NS) (**b**) columns. MP composition: CO_2_/MeOH/H_2_O (80/20/1.0, *v*/*v*/*v*); flow rate 1.00 mL min^−1^; column temperature 25 °C; BP 13.8 MPa; UV detection 254 nm (TS) and CO_2_/EtOH/PrOH/TFA/TEA (85/13/2/0.2/0.2, *v*/*v*/*v*/*v*/*v*); flow rate 1.20 mL min^−1^; column temperature 25 °C; BP 13.8 MPa; UV detection 220 nm (NS). Lines are to guide the eyes only.

**Figure 7 molecules-28-01202-f007:**
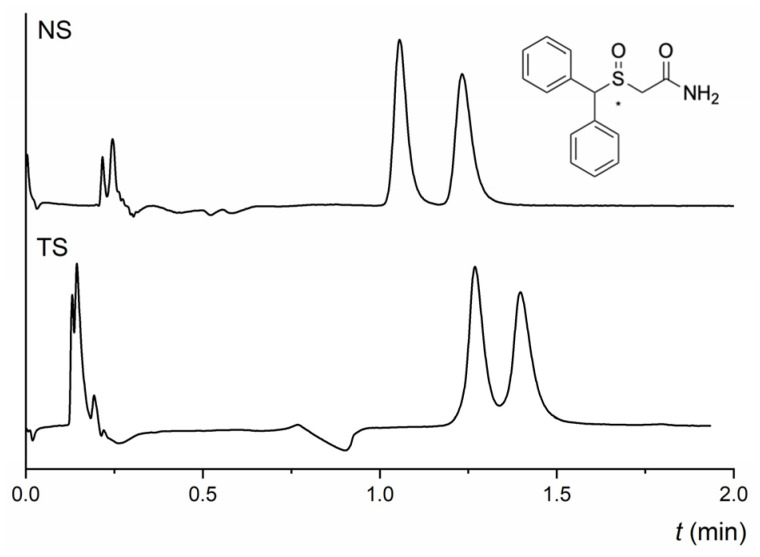
Chromatograms of fast baseline enantioseparation of modafinil on NicoShell (NS) and TeicoShell (TS) columns. MP composition: CO_2_/MeOH/H_2_O (80/20/1.25, *v*/*v*/*v*); flow rate 1.00 mL min^−1^; column temperature 25 °C; BP 13.8 MPa, UV detection 220 nm (NS) and CO_2_/EtOH/PrOH/H_2_O/TFA/TEA (85/12/3/0.5/0.1/0.1, *v*/*v*/*v*/*v*/*v*/*v*); flow rate 2.00 mL min^−1^; column temperature 25 °C; BP 13.8 MPa, UV detection 220 nm (TS). (*) in the structure indicates a stereogenic center.

**Table 1 molecules-28-01202-t001:** The best results obtained on the NicoShell and TeicoShell columns. SFC conditions: back pressure (BP) 13.8 MPa; UV detection 254 or 280 nm; injection volume 0.2–1.0 µL.

		NicoShell						TeicoShell					
Compounds	log *P*	*t*_R1_ ^a^	*R* ^b^	*α* ^c^	MP ^d^	*f* ^e^	*T* ^f^	*t*_R1_ ^a^	*R* ^b^	*α* ^c^	MP ^d^	*f* ^e^	*T* ^f^
**I** ^g^													
4-F-PV8	4.40	3.34	0.91	1.10	C/M/H_2_O (80/20/1.0)	1.00	25	1.54	1.05	1.11	C/E/P/H_2_O/TFA/TEA (85/12/3/0.25/0.2/0.2)	1.20	25
*α*-PVP	3.36	3.56	0.33	1.10	C/M/TFA/IPA (90/10/0.1/0.01)	1.00	25	2.38	0.93	1.10	C/E/P/H_2_O/TFA/TEA (85/12/3/0.25/0.2/0.2)	1.20	25
4-Cl-PVP	3.97	4.76	**1.50**	1.10	C/M/H_2_O (80/20/1.25)	1.00	25	2.20	0.84	1.10	C/E/P/H_2_O/TFA/TEA (85/12/3/0.25/0.2/0.2)	1.20	25
4-F-PVP	3.51	2.84	0.62	1.12	C/M/TFA/IPA (90/10/0.1/0.01)	1.00	25	2.00	0.87	1.10	C/E/P/TFA/TEA (85/12/3/0.2/0.2)	1.20	25
4-MeO-*α*-PVP	3.21	4.22	S ^k^	1.10	C/M/H_2_O/TFA/IPA (90/10/0.5/0.1/0.01)	1.00	25	2.66	1.40	1.12	C/E/P/H_2_O/TFA/TEA (85/12/3/0.5/0.2/0.2)	1.20	25
4-MPrC	3.88	3.65	0.67	1.11	C/M/AcA/TEA (90/10/0.2/0.02)	1.00	40	2.60	**1.53**	1.10	C/E/P/H_2_O/TFA/TEA (85/12/3/0.25/0.2/0.2)	1.20	25
*α*-PPP	2.40	1.28	**3.30**	1.34	C/M/TEA (90/10/0.1)	1.00	40	14.66	1.33	1.11	C/M/TFA/DEA (90/10/0.5/0.01)	1.00	40
M-PPP	2.91	1.44	**3.00**	1.29	C/M/TEA (90/10/0.1)	1.00	40	13.54	1.00	1.10	C/M/TFA/DEA (90/10/0.5/0.01)	1.00	40
*α*-PiHP	3.65	1.47	0.98	1.10	C/M/AcA/TEA (90/10/0.5/0.01)	1.00	40	2.24	0.66	1.10	C/E/P/TFA/TEA (85/12/3/0.2/0.2)	1.20	25
naphyrone	4.35	5.54	0.92	1.13	C/M/TFA/DEA (90/10/0.1/0.05)	1.00	40	2.96	1.03	1.10	C/E/P/H_2_O/TFA/TEA (85/12/3/0.5/0.2/0.2)	1.20	25
TH-PVP	4.82	3.65	0.67	1.11	C/M/AcA/TEA (90/10/0.2/0.02)	1.00	40	2.59	**1.50**	1.10	C/E/P/H_2_O/TFA/TEA (85/12/3/0.25/0.2/0.2)	1.20	25
PV9	4.70	2.87	0.64	1.10	C/M/TFA/IPA (90/10/0.1/0.01)	1.00	25	2.00	0.85	1.10	C/E/P/TFA/TEA (85/12/3/0.2/0.2)	1.20	25
PV10	5.14	2.80	0.53	1.10	C/M/TFA/IPA (90/10/0.1/0.01)	1.00	25	1.95	0.80	1.10	C/E/P/TFA/TEA (85/12/3/0.2/0.2)	1.20	25
5-DBFPV	3.25	6.05	0.53	1.10	C/M/AcA/TEA (90/10/0.2/0.02)	1.00	40	2.29	**1.50**	1.10	C/E/H_2_O/TFA/TEA (85/15/0.5/0.1/0.1)	2.00	25
4-M-PHP	4.32	3.00	S ^k^	1.10	C/M/TFA/IPA (90/10/0.1/0.01)	1.00	25	1.91	1.22	1.10	C/E/P/H_2_O/TFA/TEA (85/12/3/0.25/0.2/0.2)	1.20	25
3,4-MD-PHP	3.43	1.37	**1.50**	1.12	C/M/H_2_O/AcA/TEA (90/10/0.5/0.1/0.01)	1.00	25	X ^m^					
**II** ^h^													
4-CBC	3.54	6.18	**1.74**	1.11	C/M/H_2_O (80/20/1.0)	1.00	25	0.69	1.23	1.17	C/E/P/TFA/TEA (85/12/3/0.2/0.2)	1.20	25
4-CIC	2.99	3.02	0.81	1.10	C/M/H_2_O (80/20/1.0)	1.00	35	1.61	**1.85**	1.14	C/E/H_2_O/TFA/TEA (85/15/0.5/0.1/0.1)	2.00	25
4-CDC	2.60	1.33	**1.53**	1.12	C/M/H_2_O/AcA/TEA (90/10/0.5/0.1/0.01)	1.00	25	X ^m^					
**III** ^i^													
*N*-ethylketamine	3.70	1.25	**1.50**	1.18	C/M/AcA/TEA (90/10/0.5/0.01)	1.00	40	4.56	**1.64**	1.11	C/M/TFA/TEA (90/10/0.5/0.01)	1.00	40
DXE	2.74	7.98	**1.62**	1.12	C/M/H_2_O/AcA/TEA (90/10/0.5/0.1/0.01)	1.00	25	5.26	0.57	1.10	C/M/A/TFA/DEA (90/8/2/0.05/0.05)	1.00	40
2-MeO-ketamine	2.94	5.36	S ^k^	1.10	C/M/H_2_O (80/20/1.0)	1.00	35	1.01	**1.53**	1.10	C/E/H_2_O/TFA/TEA (85/15/0.5/0.1/0.1)	2.00	25
**IV** ^j^													
amphetamine	1.80	X ^m^						X ^m^					
methamphetamine	2.24	X ^m^						2.34	S ^k^	1.10	C/E/TFA/IPA (90/10/0.2/0.02)	2.00	40
citalopram	3.76	2.62	0.95	1.10	C/M/DEA (90/10/0.1)	1.00	40	X ^m^					
modafinil	1.53	1.12	**2.56**	1.25	C/M/H_2_O (80/20/1.0)	1.00	25	1.27	**1.55**	1.11	C/E/H_2_O/TFA/TEA (85/15/0.5/0.1/0.1)	2.00	25
praziquantel	2.30	0.86	0.95	1.12	C/M/AcA/TEA (90/10/0.5/0.01)	1.00	40	1.03	0.83	1.10	C/M/AcA/TEA (90/10/0.2/0.02)	1.00	40
chlorthalidone	1.60	24.78	S ^k^	1.10	C/M/AcA/IPA (90/10/0.1/0.01)	2.00	40	4.19	**1.57**	1.12	C/E/H_2_O/TFA/TEA (85/15/0.5/0.1/0.1)	2.00	25

NOTE: baseline enantioseparations are shown in bold. ^a^ Retention time of the first eluted enantiomer, ^b^ resolution, ^c^ enantioselectivity, ^d^ mobile phase composition: C—carbon dioxide, M—methanol, E—ethanol, P—propane-2-ol, A—acetonitrile, DEA—diethylamine, TEA—triethylamine, IPA—isopropylamine, AcA—acetic acid, TFA—trifluoroacetic acid, ^e^ flow rate [mL min^−1^], ^f^ temperature [°C], ^g^ derivatives of pyrovalerone, ^h^ derivatives of cathinone, ^i^ derivatives of ketamine, ^j^ others, ^k^ slight indication of enantioseparation, ^m^ no indication of enantioseparation.

## Data Availability

Data are contained within the manuscript or the Appendix A.

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
