# Peer review of "The Enantioselective Potential of NicoShell and TeicoShell Columns for Basic Pharmaceuticals and Forensic Drugs in Sub/Supercritical Fluid Chromatography"

_molecules, 2023, doi:10.3390/molecules28031202_

Round 1

Reviewer 1 Report

In this manuscript, the authors evaluate and discuss the difference in the enantioselective potential of these MG-based columns (NicoShell and TeicoShell) and their complementary behavior in SFF. Although the manuscript contains important data, the authors should address the below mentioned comments and change the manuscript accordingly prior to the final decision.

1. L.120: Figure 1 (b) describe success rate during the first screening measurements in MP composition. However, it is not necessary for this manuscript.

2. L.212, 225,226: “CH3” change to “CH3”.

Author Response

Thank you for the evaluation of the manuscript and valuable comments.

  1. L.120: Figure 1 (b) describe success rate during the first screening measurements in MP composition. However, it is not necessary for this manuscript.

Response 1: The Figure 1b has been removed from the manuscript as proposed by reviewer.

2. L.212, 225,226: “CH3” change to “CH3”.

Response 2: "CH3" has been changed to "CH3".

Several language issues have been corrected.

Reviewer 2 Report

It is a nice paper. The usefulness of chiral SFC is shown excellently. The big amounts of data well support the results and consequences. The presentations of data are little bit monotonical. The main tendencies could have been more emphasized.

Some minor critical notes:

The original meaning of SPP-based abbreviation is not mentioned.

The propanol (PrOH) is mentioned in the beginning the result and discussion section, but propan-2-ol was used in the experiments. The abbreviation list of the Table 1 is not complete, the abbreviations of several additives are missing.

The reviewer missed some explanations for the temperature effects on enatioresolution values of analytes. Are the ln alfa - 1/T curves linear?

I recommend some minor corrections.

Author Response

Thank you for the evaluation of the manuscript and valuable comments.

Point 1: The presentations of data are little bit monotonical. The main tendencies could have been more emphasized.

Response 1: We went through the paper and improved some parts of discussion to emphasize the most important parts.

Point 2: The original meaning of SPP-based abbreviation is not mentioned.

Response 2: Thank you for the comment. We have changed it to SPPs-based instead, which is already mentioned in the text. In the present form it should be clearer.

Point 3: The propanol (PrOH) is mentioned in the beginning the result and discussion section, but propan-2-ol was used in the experiments. The abbreviation list of the Table 1 is not complete, the abbreviations of several additives are missing.

Response 3: We have added the abbreviations of additives under the Table 1. All abbreviations should already be defined.

Point 4: The reviewer missed some explanations for the temperature effects on enatioresolution values of analytes. Are the ln alfa - 1/T curves linear?

Response 4: We have constructed these dependencies for selected compounds and both columns and added them to the Supplementary Material as Figures S2 and S3. We have also discussed this behavior in the text in the section 2.3 (page 8, lines 220-226). For TeicoShell column the dependency ln α on 1/T is not linear. In the case of NicoShell column the dependency is "almost" linear, there is a slight deviation from "full" linearity.

Reviewer 3 Report

The manuscript is well written and well presented and after some minor revisions it may be accepted in Molecules:

The effect of the acidic and basic additives are well performed but not well discussed. The authors should be explained the cause of these effects.

Why is the temperature limited to 40 oC? Did the authors try 50 or 60 oC?

Did the authors observe any reversal of enantiomer elution order when they changed the conditions?

Self-citations should be reduced

Author Response

Thank you for the evaluation of the manuscript and valuable comments.

Point 1: The effect of the acidic and basic additives are well performed but not well discussed. The authors should be explained the cause of these effects.

Response 1: We have included some paragraphs concerning effect of acidic and basic additives in section 2.2 Effect of additives (page 6, lines 156-160, 162-171).

Point 2: Why is the temperature limited to 40 oC? Did the authors try 50 or 60 oC?

Response 2:  We did not tested these temperatures because they are  above the temperature limit recommended by producer.

Point 3: Did the authors observe any reversal of enantiomer elution order when they changed the conditions?

Response 3: Unfortunatelly, we do not have individual enantiomers of compounds of interest. Therefore we did not check enantiomer elution order.

Point 4: Self-citations should be reduced

Response 4: Self-citations have been reduced. The references 19 and 32 (original version of manuscript) which were not essential were removed.